# Motor Learning of Complex Tasks with Augmented Feedback: Modality-Dependent Effectiveness

**DOI:** 10.3390/ijerph182312495

**Published:** 2021-11-27

**Authors:** Jarosław Jaszczur-Nowicki, Oscar Romero-Ramos, Łukasz Rydzik, Tadeusz Ambroży, Michał Biegajło, Marta Nogal, Waldemar Wiśniowski, Dariusz Kruczkowski, Iwona Łuszczewska-Sierakowska, Tomasz Niźnikowski

**Affiliations:** 1Department of Tourism, Recreation and Ecology, University of Warmia and Mazury in Olsztyn, 10-719 Olsztyn, Poland; j.jaszczur-nowicki@uwm.edu.pl; 2Department of Didactics of Languages, Arts and Sports, University of Malaga, 4, 29017 Málaga, Spain; oromero@uma.es; 3Institute of Sports Sciences, University of Physical Education, 31-571 Krakow, Poland; tadek@ambrozy.pl; 4Faculty of Physical Education and Health, Józef Piłsudski University of Physical Education, 21-500 Biala Podlaska, Poland; michal.biegajlo@awf.edu.pl (M.B.); marta.nogal@awf.edu.pl (M.N.); waldemar.wisniowski@awf.edu.pl (W.W.); 5Faculty of Health Sciences, Elbląg University of the Humanities and Economics, 82-300 Elbląg, Poland; dyrektor@olimpijczyk.gda.pl; 6Department of Human Anatomy, Medical University of Lublin, 20-090 Lublin, Poland; iwona.ikona@gmail.com

**Keywords:** training sessions, verbal, visual, verbal–visual feedback, vertical jump, complex task

## Abstract

Background: This paper aims to evaluate the effectiveness of feedback modalities in the motor learning of complex tasks. Methods: This study examined sixty-one male university students randomised to three groups: group Verbal (VER) = 20 (body height 178.6 ± 4.3 cm, body mass 81.3 ± 3.7 kg, age 20.3 ± 1.2 years), group Visual (VIS) = 21 (body height 179 ± 4.6 cm, body mass 82 ± 3.4 kg, age 20.3 ± 1.2 years), and group Verbal–Visual (VER&VIS) = 20 (body height 178.6 ± 4.3 cm, body mass 81.3 ± 3.7 kg, age 20.3 ± 1.2 years). The duration of the experiment was 6 months. Training sessions were performed three times per week (on Mondays, Wednesdays, and Fridays). The participants were instructed to perform a vertical jump with an arm swing (with forward and upward motion). During the jump, the participants pulled their knees up to their chests and grabbed their lower legs. The jump was completed with a half-squat landing, with arms positioned sideward. The jumping performance was rated by three gymnastic judges on a scale from 1 to 10. Results: A Tukey post hoc test revealed that in the post-test, a significant difference in the quality of performance was found between the Verbal group concerning errors combined with visual feedback on how to correct them (VER&VIS), the Verbal group concerning errors (VER), and the Visual group with visual feedback on the correctness of task performance (VIS). The ratings observed in the post-test were significantly higher in group VER&VIS than in groups VER and VIS (9%; *p* < 0.01 and 15%; *p* < 0.001, respectively). All judges’ ratings observed in group VER&VIS and VIS decreased insignificantly, but in group VER the ratings improved insignificantly. Conclusion: Providing verbal feedback combined with visual feedback on how to correct errors made in performing vertical jumps proved more effective than the provision of verbal feedback only or visual feedback only.

## 1. Introduction

Numerous researchers have attempted to determine how the process of learning motor tasks is affected by different variables, such as the type of feedback given to the learner (time, type, frequency) [1,2], training organisation (task complexity, context effects) [3,4], training type (physical, mental) and different versions of feedback (verbal, kinaesthetic, visual) [5,6,7,8,9]. So far, simple motor tasks that can be learnt without needing numerous attempts have been used in these studies [3,4]. Numerous advantages of implementing simple tasks (objectivity of measuring effects and time effectiveness) have been emphasised [5,6]. PE teachers, coaches, and physiotherapists should attempt to understand the general guidelines for learning complex motor tasks to improve the breadth of evidence-based practical tips they give during PE lessons or training. This can be accomplished through research using complex motor tasks that are more demanding for learners in terms of cognition and fitness [1,10,11,12,13,14,15,16].

In studies on observational learning, it was noted that video demonstration did not lead to an improvement in learning outcomes if it was not combined with verbal instruction. This phenomenon mainly relates to complex motor tasks [17]. Researchers claim that in order to enhance the effects of video demonstration, it needs to be combined with verbal feedback [18,19] or, additionally, light reflective markers or a model with light reflective markers on the most important body segments should be used. Hebert and Landin [20] believe that providing visual feedback combined with verbal feedback makes it possible to draw learners’ attention to key elements of a motor task. According to Tzetzis et al. [21], the effectiveness of video demonstration depends on the complexity of a task. When examining the performance of complex motor tasks, Sadowski, Mastalerz, and Niźnikowski [22] noted that giving detailed feedback on key technical elements may exert a more beneficial influence on the process of learning than giving 100% feedback on all task-related errors. The authors claim that more experienced athletes may also make use of intrinsic feedback and in this way they can achieve their goal. In the case of learning complex tasks, determining and providing bandwidth feedback may be more beneficial for experienced learners who possess certain motor skills.

Information regarding effective feedback is particularly important both for PE teachers and coaches [23]. In their study, Nogal and Niźnikowski [16] confirmed that well-adjusted feedback was a key factor in learning complex motor tasks. Lee et al. [24] stated that positive verbal feedback is useful in non-specific tasks. Kernodle and Carlton [25] pointed out providing that verbal feedback on errors and on how to correct them is particularly helpful for beginners who, when given this type of information, can not only improve their skills but also gain self-confidence. Similarly, Smith and Davies [26] noted that receiving feedback on errors and on how to correct them may exert a considerable influence on athletes’ mentality and self-confidence.

Coaches, instructors, PE teachers, and physiotherapists ought to know if the effectiveness of learning new motor tasks will increase when the learner practises in conditions that require extra cognitive effort and activity connected with processing information through delaying feedback, limiting cues, or accepting some errors.

The aim of this study is to determine in what way verbal, visual, and verbal–visual feedback influences the effectiveness of learning identical motor tasks. If this problem is not solved, it will not be possible to identify effective rules for learning tasks with different structures of movement. The work of coaches, PE teachers, and physiotherapists will thus continue to be based on intuition rather than scientific knowledge. This problem will hamper the optimisation of the training process (especially in technical training) and discourage students from participating in PE lessons and mastering more and more difficult motor tasks.

## 2. Materials and Methods

The participants were sixty-one male university students who did not play sports. Participants were randomised into three groups: group Verbal (VER) = 20 (body height 178.6 ± 4.3 cm, body mass 81.3 ± 3.7 kg, age 20.3 ± 1.2 years), group Visual (VIS) = 21 (body height 179 ± 4.6 cm, body mass 82 ± 3.4 kg, age 20.3 ± 1.2 years), and group Verbal–Visual (VER&VIS) = 20 (body height 178.6 ± 4.3 cm, body mass 81.3 ± 3.7 kg, age 20.3 ± 1.2 years).

Research design: The experiment lasted 6 weeks, with training sessions performed 3 times per week (on Mondays, Wednesdays, and Fridays). Each student participated in 18 sessions lasting 60 min, and during the workout each subject performed 20 exercises, which were divided into 4 sets with 5 repetitions each (approximately 3 min per person). The participants learnt how to perform a vertical jump with an arm swing. During the jump, they pulled their knees to their chest and were instructed to grab their lower legs, and then perform a half-squat landing with arms sidewards (VJPKL). The participants had not performed similar tasks prior to the experiment. The progressive part method was used, with the task divided into individual parts. The students learnt the preparatory phase during sessions 1 to 4, the main phase during sessions 5 to 8 and the final phase in sessions 9 to 12. Eventually, during sessions 13 to 16, the participants learnt the entire movement task. During each training session, 20 exercises were divided into 4 sets with 5 repetitions. Each set was followed by feedback. Group VER received verbal feedback on errors. Group VIS received visual feedback (concerning the correctness of performance). Group VER&VIS received verbal feedback on errors and visual feedback on how to correct them. Two days prior to the experiment, the participants performed a pre-test (PRET), and a post-test (POST) was conducted one day after it ended. Furthermore, a retention test (RETT) was carried out seven days after the experiment. The participants were instructed to follow a standardised warm-up and to perform a single movement task. Their performance was rated by 3 gymnastic judges on a scale from 1 to 10 based on the rules of the International Gymnastics Federation. During the evaluation, 0 to 0.3 points were deducted from a maximum score of 10 points for each minor error, 0.4 to 0.6 points for medium errors, and 0.7 to 1 points for major errors. The concordance of the experts’ ratings was verified using the concordance coefficient (0.94). All the participants were informed in a written and oral manner about the tests that would be carried out in the study and they signed the informed consent form. The consent of the Senate Committee on Research Ethics of the University of Physical Education in Warsaw was obtained.

Statistical analysis: The ANOVA analysis of variance was employed to evaluate the statistical significance of differences between the measurements. The Shapiro–Wilk test was used to test the data for normality of distribution and homogeneity of variances. After the prerequisite was verified, the variables were analysed using a two-way mixed-factor ANOVA, Group (3) × Test Time (3) for judge ratings and Group (3) × Test Time (2) for force measurements. The three experimental groups represented a between-subjects factor, whereas the testing times were considered a within-subjects factor. A probability level was considered critical at *p* < 0.05. The Tukey post-hoc test was used to examine significant differences. The statistical analysis of the results was performed using the Statistica software (StatSoft Inc. (2017), STATISTICA data analysis software system, version 13.3, TIBCO Software Inc., Palo Alto, CA, USA, www.statsoft.com) (accessed on 16 August 2021).

## 3. Results

The repeated-measures ANOVA revealed significant effects of Group (F(2,54) = 11.69; *p* = 0.001) and Group × Test Time interaction (F(4,108) = 40.52; *p* = 0.001). Figure 1 presents the means and standard deviations.

Based on a Tukey post-hoc test, the significance of differences between mean ratings of the judges at particular stages of the learning process was determined for groups VER&VIS, VIS and VER (Table 1).

In the study, it was found that in the pre-test, differences in mean scores of the groups for VJPKL performance were not statistically significant F(2,54) = 1.344, *p* < 0.269). However, after 6 weeks (post-test), these differences were significant F(2,54) = 38.651, *p* < 0.001). The mean score of group VER&VIS was better than the mean scores of groups VIS and VER by 9% (*p* < 0.01) and 15% (*p* < 0.001), respectively. In turn, group VIS had a better mean score than group VER, and the difference was 10% (*p* < 0.001). The retention test revealed that differences in mean scores of the groups were significant F(2,54) = 15.631, *p* < 0.001) in favour of group VER&VIS. This group had a better score than groups VIS and VER by 10% and 7%, respectively.

## 4. Discussion

Despite the numerous studies conducted so far, the issue of teaching and learning motor tasks, especially those with a complex structure of movement, needs to be addressed in terms of optimisation. For instance, during a PE lesson or training session, when learners or athletes who try to acquire new motor skills repeat the same error, teachers or coaches may choose not to give any feedback on when the error was made and how to eliminate it. When learners have experience, they will certainly use intrinsic feedback, which helps them acquire technical skills and shorten the learning process. Considerably worse effects will be produced when learners are not experienced enough and they are still incapable of using intrinsic feedback. In such cases, feedback provided by teachers or coaches is particularly meaningful.

Hence, the key question each teacher or coach should ask is ‘How can I facilitate the process of teaching and learning motor tasks and how can I optimise PE lessons or training aimed at acquiring technical skills?’. Some of the problems faced are the choice of content and the type and amount of feedback that ought to be provided in the course of the process of learning motor tasks. Should feedback be given in verbal or visual form or as a combination of both? These and other questions cannot be answered unequivocally [1,23,25,27,28,29,30,31,32]. Despite various studies on this subject, there is no empirical evidence of the influence of different types and content of feedback on the effectiveness of motor learning of complex tasks [9,13,14,15,29,33,34,35,36]. Coaches, teachers, and physiotherapists should always choose feedback adequate for learners’ levels of advancement and skills. Feedback should be simple, easy to understand, and concise so that athletes and students can receive, process, and use it when developing motor habits both now and in the future, when more complex and difficult tasks will have to be learnt. This paper attempts to evaluate the effectiveness of feedback modalities in the motor learning of complex tasks.

The team of researchers assumed that providing verbal feedback on errors combined with visual feedback on the correct performance of the task can be more efficient than just verbal feedback on errors or just visual feedback on correctness provided in isolation. The results of the study lead to the conclusion that the use of different strategies to provide feedback to learners can produce different effects. It was found that the three study groups showed statistically significant improvements in the post-test. However, in the retention test, the improvements were reported only in group VIS. The study participants from group VER&VIS, who received verbal feedback on errors and visual feedback on how to correct them, had better scores than those from VIS, who only received visual feedback regarding the correctness of task performance, and the learners from group VER, who only received verbal feedback on errors (the effect size for the post-test was 8.352 in VER&VIS, 3.894 in VIS and 8.147 in VER). Insignificantly improved scores in the retention test were found in group VER. In groups VER&VIS and VIS, the results of the retention test did reveal changes. These findings are partly consistent with data reported by studies that have tested the guidance hypothesis in complex movement skills. The authors of these studies demonstrated that after cessation of 100% feedback, the results of the retention test were worse. In our study, group VIS showed improved performance in the retention test. However, the VER&VIS group, who received more feedback, showed relatively better performance than the participants from groups VIS and VER, who were given less information.

Schmidt and Wrisberg [37] noted that proper feedback produced better effects in the process of learning motor tasks. These results are consistent with the findings of other researchers [19,38,39]. Researchers claim that the combination of verbal feedback with other sources of information increases learning effectiveness. Similarly, Weiss and Klint [40] stated that the presentation of a given task enhanced with verbal feedback improves learning effectiveness. Moreover, numerous studies indicate that there are a lot of relevant factors in the way verbal and visual feedback is provided that impact the learning process [20,39,41,42,43,44]. The knowledge of which types of feedback should be combined and in what manner forms the basis of learning process optimisation. This may indicate that providing verbal feedback on errors and visual feedback on task performance is justified when it comes to improving motor skills, while visual feedback combined with verbal feedback proves to be more effective during the learning process. Another important piece of information for both teachers and coaches is that visual feedback on performance correctness, similar to verbal feedback on errors, produces marginal learning outcomes. Considering the above facts, it may seem to be more beneficial to use visual feedback on performance correctness together with verbal feedback on errors.

The type of task performed appears to be a key determinant in the choice of strategy for providing feedback concerning the quality and results in the performance of a task. The results are consistent with those published by Tzetzis and Votsis [30], who argued that during the learning of complex tasks, positive feedback must combine information on errors and ways of correcting the task. In our study, the participants from the group VER&VIS, who received such information, showed significant improvement in the scores (*p* < 0.05). The participants from the groups VIS and VER, who received less information on task performance, obtained worse results. This indicates that the amount of information provided to a learner is the main determinant of the effectiveness of learning a complex movement task. Convincing evidence was provided by Laguna [31], who demonstrated that the effectiveness of learning complex tasks depended on the level of difficulty of the task and specificity of the feedback (task-related information). This is indirectly in line with our findings.

It seems that different types of feedback may be used depending on the type of problem that we want to solve and on the type of the process we will apply when learning. Teachers who provide learners with direct visual feedback on the correctness of task performance as well as verbal feedback on how to correct errors may expect their learners to follow these sources of information, which may considerably enhance the process of learning at an early stage. On the other hand, teachers can select a version of the learning process which, at an early stage, will pose a certain problem or will compel learners to go beyond the feedback received and to discover and develop their own solutions. Although at an early stage this approach does not produce considerable effects, it seems beneficial in the long term. Therefore, the process of learning should focus not only on speed and the ease of acquiring new motor skills, but also on encouraging learners to self-develop internally. Such an approach may be extremely important due to the fact that it will teach learners to be persistent in developing their own skills.

### Limitation of the Study

One limitation of our research is that the three learning strategies were used only for one complex movement task. It is recommended that future research should be carried out with consideration of movement tasks with different levels of complexity and different feedback strategies.

## 5. Conclusions

At the early stages of learning, a combination of visual and verbal feedback represents the basis for learning complex movement tasks and most often enhances the learning process.In the post-test, visual feedback on the correctness of task performance combined with verbal feedback on errors in task performance proved to be more effective than visual feedback only on the correctness of task performance or verbal feedback only on errors in performing a vertical jump where the participants pulled their knees up to their chest and grabbed their lower legs. In the retention test, an improvement was only noted in the case of verbal feedback on errors in task performance.Examining such variables can help us to assess the role of feedback and learning complex movement tasks.Future research should focus on verifying whether our results can be applied in learning other more complex movement tasks. For this purpose, modern methods for the verification of research results and experimental programs should be used.

### Practical Implication

The results of our research will help coaches to assess the role of feedback in learning complex movement tasks that are commonly taught by personal trainers and physical educators. Providing a lot of information gives good results in the short term, but providing limited information seems to be more effective in the long term.

## Figures and Tables

**Figure 1 ijerph-18-12495-f001:**
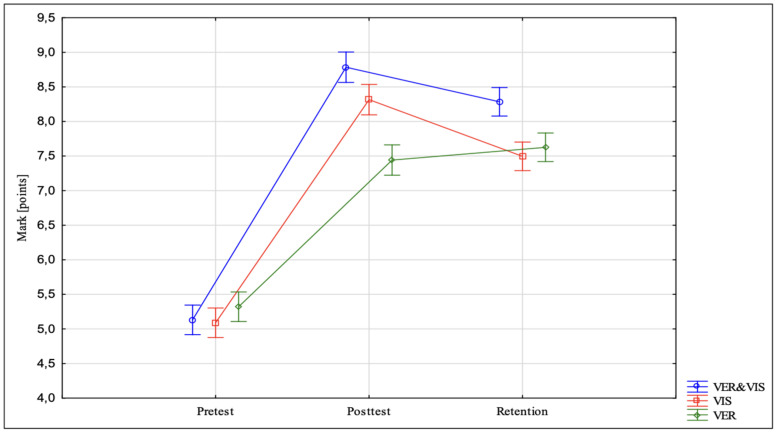
Point-based evaluation of motor task performance in all the groups with regard to time and type of feedback that influences the effectiveness of motor task learning.

**Table 1 ijerph-18-12495-t001:** Significance level of Tukey post-hoc test of experts’ ratings (PRET, POST, and RETT) in groups VER, VIS, and VER&VIS.

No.	Groups	VER&VIS (PRET)	VER&VIS (POST)	VER&VIS (RETT)	VIS (PRET)	VIS (POST)	VIS (RETT)	VER (PRET)	VER (POST)	VER (RETT)
1	VER&VIS (PRET)		0.00	0.00	0.99	0.00	0.00	0.94	0.00	0.00
2	VER&VIS (PRET)	0.00		0.00	0.00	0.06	0.00	0.00	0.00	0.00
3	VER&VIS (PRET)	0.00	0.00		0.00	1.00	0.00	0.00	0.00	0.00
4	VIS (PRET)	0.99	0.00	0.00		0.00	0.00	0.83	0.00	0.00
5	VIS (POST)	0.00	0.06	1.00	0.00		0.00	0.00	0.00	0.00
6	VIS (RETT)	0.00	0.00	0.00	0.00	0.00		0.00	0.99	0.99
7	VER (PRET)	0.94	0.00	0.00	0.83	0.00	0.00		0.00	0.00
8	VER (PRET)	0.00	0.00	0.00	0.00	0.00	0.99	0.00		0.59
9	VER (PRET)	0.00	0.00	0.00	0.00	0.00	0.99	0.00	0.59	

## Data Availability

The data presented in this study are available on request from the corresponding author.

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
