# Peer review of "Motor Learning of Complex Tasks with Augmented Feedback: Modality-Dependent Effectiveness"

_ijerph, 2021, doi:10.3390/ijerph182312495_

Round 1
Reviewer 1 Report
Dear Authors,
Thank you for the paper. Please find my comments below.
1- The paper is not novel, there are many modern ways in which performance can be improved, based on the method used in this study, which has been applied many times. For example the use of hypermedia!
2- In Table 2, please provide the data in 2 decimals.
3- How did you determine the sample? How is its size determined? Did you use any statistical methods for this?
4- I would recommend more experimental processes to improve the paper.
Author Response
Dear Reviewer,
Thank you very much for your time and valuable comments, which all have been considered and incorporated. The detailed list of responses is given below. We hope that the modifications and explanation will be acceptable for you.
Yours sincerely,
Rydzik, corresponding author
1.The paper is not novel, there are many modern ways in which performance can be improved, based on the method used in this study, which has been applied many times. For example the use of hypermedia!
A: Thank you for your attention, this is one of a series of articles using this method. We will take this technology into consideration for future research.
2.In Table 2, please provide the data in 2 decimals.
A: This has been correct
3- How did you determine the sample? How is its size determined? Did you use any statistical methods for this?
A: A specialized sampling calculator was used. Additionally, information was added in the limitation of the study
4-I would recommend more experimental processes to improve the paper.
A: Thank you for your comment, which we will include in future studies
Reviewer 2 Report
There are 61 college-age males as test subjects but there is no indication on the ability level of these persons. Were any athletes, former athletes, sedentary, resistance trained, etc.?
The test exercise appears to be a "tuck jump" and how many performed this exercise during training.
How is the information from this study useful to a personal trainer, physical education teacher (LTAD), sport coach, or strength & conditioning coach?
Merge sentences in the discussion section so at least three sentences form a paragraph.
Does the motor learning for an 10 year-old the same as a 20 year-old college athlete?
The methods needs clarification on the 18 sessions, as it reads that each session was 60 minutes in length which is a long amount of time to perform only 20 repetitions.
The authors describe experience that matters but what type of experience specifically, as an college athlete can have sport experience but lack S&C training experience.
Sentences beginning with the term "It" is not always clear what is being referred to. Please address this throughout the manuscript.
There is no mention of the type of cuing used, was internal or external cues used by the instructors?
The application of the information is vague and needs to be clarified so practitioners can effectively use the results of the study.
Author Response
Dear Reviewer,
Thank you very much for your time and valuable comments, which all have been considered and incorporated. The detailed list of responses is given below. We hope that the modifications and explanation will be acceptable for you.
Yours sincerely,
Rydzik, corresponding author
There are 61 college-age males as test subjects but there is no indication on the ability level of these persons. Were any athletes, former athletes, sedentary, resistance trained, etc.?
A: There were 61 college-aged students, none of which had a history of participation in organized sports or resistance training.
The test exercise appears to be a "tuck jump" and how many performed this exercise during training.
A: Each participant completed a total of 20 repetitions during the training session. Specifically, each participant completed four sets of five repetitions. Each set was separated by approximately 30 seconds.
How is the information from this study useful to a personal trainer, physical education teacher (LTAD), sport coach, or strength & conditioning coach?
A: Examining such variables can help assess the role of feedback and learning complex movement tasks, which are commonly taught by personal trainers and physical educators. Using a lot of information gives good results in the short term, but providing limited information seems to be more effective in the long term.
Merge sentences in the discussion section so at least three sentences form a paragraph.
A: This has been correct
Does the motor learning for an 10 year-old the same as a 20 year-old college athlete?
A: The motor learning and motor development literature suggests that the motor learning between these to populations is different.
The methods needs clarification on the 18 sessions, as it reads that each session was 60 minutes in length which is a long amount of time to perform only 20 repetitions.
A: The entire workout for a given group lasted 60 min, and during the workout each subject performed twenty repetitions were divided into four sets with 5 repetitions (approximately 3 minutes per person). In other words, the entire group was active for approximately 60 minutes, but the individual assessment to collect the dependent variables for the present study lasted approximately three minutes per participant.
The authors describe experience that matters but what type of experience specifically, as an college athlete can have sport experience but lack S&C training experience.
A: There were 61 college-aged students, none of which had a history of participation in organized sports or resistance training.
Sentences beginning with the term "It" is not always clear what is being referred to. Please address this throughout the manuscript.
A: We agree with the reviewer and have made corrections.
There is no mention of the type of cuing used, was internal or external cues used by the instructors?
A: The instructors used neutral cues and description and prescription feedback. It was not a goal of the study to direct attention internally or externally.
The application of the information is vague and needs to be clarified so practitioners can effectively use the results of the study.
A: Added practical implication,
Examples of feedback: verbal feedback on errors:
– jump was performed leaning too far forward
– jump was performed leaning too far backwards
– you drew your knees to your chest at the beginning of the jump
– you drew your knees to your chest at the end of the jump
– hands grabbed the shins at the beginning of the jump
– hands grabbed the shins at the end of the jump
– tucking done at the beginning of the jump
– tucking done at the end of the jump
– untucking done at the beginning of the jump
– untucking done at the end of the jump
- your knee were straight at the landing
- your knees and hips were straight at the landing
- you kept your arms to the side
Reviewer 3 Report
GENERAL COMMENTS
The topic of the paper is interesting and fits the scope of the journal. The text is relatively well written and composed. However I am not sure if this paper offers additional information. There are many papers in the literature with the same issue.
SPECIFIC COMMENTS
Why dd you choose this test?
Please rewrite the purpose of this study.
Lines 193-194. Please explain why do you believe that the results of two groups in the retention test were deteriorated?
Author Response
Dear Reviewer,
Thank you very much for your time and valuable comments, which all have been considered and incorporated. The detailed list of responses is given below. We hope that the modifications and explanation will be acceptable for you.
Yours sincerely,
Rydzik, corresponding author
GENERAL COMMENTS
The topic of the paper is interesting and fits the scope of the journal. The text is relatively well written and composed. However I am not sure if this paper offers additional information. There are many papers in the literature with the same issue.
A: Thank you for reviewing our manuscript. The scientific novelty of our work is the identification of the roles of feedback during the learning of complex motor tasks.
SPECIFIC COMMENTS
Why did you choose this test?
A: This test was chosen because we have previously conducted research using this test, we feel expert in its implementation, and we plan to use other tests in future work.
Please rewrite the purpose of this study.
A: This has been correct
Lines 193-194. Please explain why do you believe that the results of two groups in the retention test were deteriorated?
A: This has been correct
Round 2
Reviewer 1 Report
Dear Authors,
Thank you for your submission.
Please, find my comments below:
The authors did not use modern methods and the paper. The methods are too old. I asked them to improve the methods but they didn't do it.
The paper did not offer additional information based on others that have been published on this topic.
I asked them how did you determine the sample?
The answer must be added in the paper in detail but they said: A: A specialized sampling calculator was used.
Finally, 9 authors in the 1st version and 1 additional in the 2nd version for a very old methodology!! it seems unsuitable for the journal and unreliable for the reader as well.
Author Response
Dear Reviewer,
Thank you for your comments, which we have taken into account. Please see below for detailed responses. We were surprised by your approach because you didn't have many comments in the first round, and those that were, we tried to do a thorough job.
Your Sincerly,
Rydzik- Correspondning Author
Dear Authors,
Thank you for your submission.
Please, find my comments below:
The authors did not use modern methods and the paper. The methods are too old. I asked them to improve the methods but they didn't do it.
A:We respectfully disagree with the reviewer. In all gymnastic competitions (e.x. the Olympics) judges are used to evaluate the performance level of athletes. We adopted a similar method in the present study to evaluate our participants. We feel the method used in our study is very modern and adds great face validity to our findings and results.
The paper did not offer additional information based on others that have been published on this topic.
A:We have made modifications and now the paper does offer additional information based on previous work which has been published on this topic: „PE teachers, coaches or physiotherapists should gain an understanding of the general guidelines of learning complex motor tasks to improve their breadth of evidence-based practical tips for use during PE lessons or training, which is accomplished through research using complex motor tasks that are more demanding for learners in terms of cognition and fitness [1,10–16]."
I asked them how did you determine the sample?
The answer must be added in the paper in detail but they said: A: A specialized sampling calculator was used.
A: Power analysis of the research using G*Power Version 3.1.9.4 (Faul et al., 2007) showed that with estimated moderate effect size, it was determined that a minimum of ten participants were required in each group (effect size f = 0.60, power = 0.95, p = 0.05). Therefore, the recruited sample of 10 participants in each group was considered appropriate.
Finally, 9 authors in the 1st version and 1 additional in the 2nd version for a very old methodology!! it seems unsuitable for the journal and unreliable for the reader as well.
A: The additional author helped to refine the paper and to fund it. His contribution to the review process was important enough that we included him in the authorship.
Reviewer 2 Report
Rewrite this sentence in the introduction for clarity, suggestion; "PE teachers, coaches or physiotherapists should gain an understanding of the general guidelines of learning complex motor tasks to improve their breadth of evidence-based practical tips for use during PE lessons or training, which is accomplished through research using complex motor tasks that are more demanding for learners in terms of cognition and fitness [1,10–16]."
There are fragmented paragraphs in the discussion section as there are single sentences that need merging into other ones.
A table that presents the stages of development and suggestions for motor task feedback would benefit the practitioners that work at different levels.
Improvements have made reading and understanding the document more clearly.
Author Response
Dear Reviewer,
Thank you very much for your time and valuable comments, which all have been considered and incorporated. The detailed list of responses is given below. We hope that the modifications and explanation will be acceptable for you.
Yours sincerely,
Rydzik, corresponding author
Rewrite this sentence in the introduction for clarity, suggestion; "PE teachers, coaches or physiotherapists should gain an understanding of the general guidelines of learning complex motor tasks to improve their breadth of evidence-based practical tips for use during PE lessons or training, which is accomplished through research using complex motor tasks that are more demanding for learners in terms of cognition and fitness [1,10–16]."
A: This has be correct
There are fragmented paragraphs in the discussion section as there are single sentences that need merging into other ones.
A: This has be correct
A table that presents the stages of development and suggestions for motor task feedback would benefit the practitioners that work at different levels.
A: According to the authors, inserting a table is not necessary in this study
Improvements have made reading and understanding the document more clearly.
A: Thank you
Reviewer 3 Report
Dear authors, paper accepted.
Author Response
Thank you